# The Impact of China's Grassland Ecological Compensation Policy on the Income Gap between Herder Households? A Case Study from a Typical Pilot Area

Zhidong Li [1,2] , Didi Rao [1,2] and Moucheng Liu [1,*]

1   Institute of Geographic Sciences and Natural Resources Research, Chinese Academy of Sciences, Beijing 100101, China; lizd.18s@igsnrr.ac.cn (Z.L.); raodd.19b@igsnrr.ac.cn (D.R.)
2   University of Chinese Academy of Sciences, Beijing 100049, China
*   Correspondence: liumc@igsnrr.ac.cn

**Abstract:** China's policy of subsidies and rewards for grassland ecological protection (PSRGEP) aims to maintain the ecological function of grasslands and increase the income of herder households. Since 2011, the Chinese government has invested more than 150 billion yuan in this policy, making it currently the largest grassland ecological compensation project in China. Based on a survey of 203 herder households in Xin Barag Left Banner, Inner Mongolia Autonomous Region, this study used the Lorenz curve and Gini index to describe the imbalance in the distribution of compensation funds. Then, the integrated livelihood capital scores before compensation were used as a baseline. The changes in ranking and standard deviation of the scores after receiving compensation funds were analysed to draw a conclusion about the impact on the income gap between herder households. Finally, we described the absolute income gap through a group comparison. The results show that the distribution of compensation funds is unbalanced (Gini index is 0.46). According to the order of compensation funds from high to low, the top 20% of sample herder households received 49% of the total funds. Given the unbalanced distribution, households with better family economic conditions received more compensation funds. After receiving the compensation funds, the change in the ranking of the household's livelihood capital integrated score was small, but the standard deviation increased from 0.1697 to 0.1734, and the Gini index of the households' capital integrated scores decreased from 0.35 to 0.34 (the coefficient of variation decreased from 0.66 to 0.63). The group with the highest integrated livelihood capital score received 3.6 times the compensation funds of the group with the lowest score. As a result, under the promotion of PSRGEP, the local absolute income gap has widened, but the relative income gap has reduced. This study evaluated the current distribution of compensation funds for PSRGEP, which could provide a scientific basis for managers to optimize the fund distribution in the future.

**Keywords:** payments for ecosystem services; policy of subsidies and rewards for grassland ecological protection; livelihood capital; Gini index

## 1. Introduction

Grassland is one of the largest terrestrial ecosystems and an important source of food and energy for humans [1–3]. China is rich in grassland resources [4–7]. Grasslands have important ecological functions, covering an area of approximately 30–40% of the country's total land area [8–10]. However, China's grasslands have been seriously degraded in recent years. Relevant studies have shown that human activities, such as overgrazing, are among the main driving forces leading to grassland degradation [11–14]. To prevent the continued destruction of the grassland ecological environment by human activities, the Chinese government implemented the policy of subsidies and rewards for grassland ecological protection (PSRGEP) in 2011.

The idea of the PSRGEP is that the government provides subsidies and rewards (compensation funds) for herder households in the pastoral areas of 13 provinces to encourage them to reduce their grazing activities. In this way, the production intensity of the grassland will decrease, and the ecological function of the grassland will gradually recover. According to the regulations of the Chinese central government, the grassland in the pilot area is divided into no-grazing areas and forage–livestock balance areas. Grazing is prohibited in the no-grazing areas, and the annual subsidy is 112.5 yuan/ha. In the forage–livestock balance areas, the number of livestock is controlled according to the quality of the pasture, and the herder households who meet the requirements are rewarded 37.5 yuan/ha per year. Each pilot province can appropriately adjust the amount according to the actual situation.

As an effective measure to protect the ecological environment, ecological compensation is receiving extensive attention worldwide [15–19]. At this stage, the Chinese government has implemented ecological compensation practices in watersheds [20], cultivated land [21–23], protected areas [24] and other fields. As the most vital component of China's grassland ecological compensation program, the PSRGEP involves a large transfer payment (the Chinese government has allocated more than 150 billion yuan in 10 years), a large number of people, and a large coverage area. These factors have drawn scholars' attention to this policy. Studies have expounded on the ecological effects of the PSRGEP through remote sensing data and grassland monitoring data. The results show that since the implementation of the policy, the quality of grassland in China has improved significantly [25,26]. However, other studies have focused on changes in the livelihood of herder households in the context of the PSRGEP effects, for example, the incentives for the reduction in herder households' livestock [27] and the transformation of herder households' production methods [28]. One of the most concerning factors is the impact of the PSRGEP on households' income. Changes in income can easily affect the livelihood decisions of herder households, and it is critical to the outcome and continuity of the policy.

Regarding the impact on income, the PSRGEP has increased the total income and the proportion of non-agricultural income of herder households, but herder households are still unable to eliminate their dependence on traditional animal husbandry [29]. With the increase in the compensation standard, the income level of herder households has also increased [30]. However, the compensation level of the PSRGEP is still not enough to cover the extra efforts made by herder households, nor does it meet the livelihood expectations of most herder households [31]. Studies have assessed the changes in herder households' income due to the policy, but there is no horizontal comparison among herders. Since the PSRGEP has increased the income of herder households, is there an imbalance in the fairness of the policy? In other words, does the PSRGEP also widen the income gap between herder households? There is almost no research on this issue. China's income gap is currently a topic of great interest to many scholars [32,33]. Most of the PSRGEP pilot programs are in the economically less-developed provinces of China, which include four ethnic minority autonomous regions (China has a total of five ethnic minority autonomous regions). Economic growth, environmental protection and social stability in these regions are vital to the development of China. However, the existence of an income gap has a negative impact [34–36]. Based on the issues described above, the "Decision of the Communist Party of China, Central Committee and the State Council on winning the battle against poverty" promulgated in 2015 pointed out that ecological compensation is a new method of poverty alleviation. According to the "Opinions on Improving the Compensation Mechanism for Ecological Protection" promulgated by the State Council of the People's Republic of China in 2016, ecological compensation funds should be oriented towards areas of poverty and poor people. Therefore, China's ecological compensation policy should function as a means of reducing the income gap. This study will explore the impact of PSRGEP on the income gap to provide an important reference for the follow-up regulation of the policy.

## 2. Materials and Methods

### 2.1. Study Site

Xin Barag Left Banner is located in southwestern Hulunbuir City, Inner Mongolia Autonomous Region, at 117°33′–120°12′ E and 47°10′–49°47′ N (Figure 1). The elevation gradually decreases from southeast to northwest, with an altitude of 560–1570 metres. The total area of the whole banner is $2.22 \times 10^4$ km$^2$, and there are 7 towns with a population of 41,813. The selection of this area as a study site was based on the following three reasons: (1) Xin Barag Left Banner is located in the agricultural and pastoral transition zone in northern China and it is one of China's important ecological barriers. In the context of China's grassland degradation, the effects of grassland ecological protection policies in this area are of general concern. (2) This is a typical area for studying the PSRGEP. Animal husbandry is the main industry of Xin Barag Left Banner, and grassland nomadism is the most common mode of production. According to the regulations of the second round of the PSRGEP, 1.45 million hectares of natural grassland are included in the compensation scope, and the total compensation funds exceed 150 million yuan. (3) Xin Barag Left Banner is an area inhabited by ethnic minorities, with Mongolians accounting for 80.2% of the total population. It is also located at the junction of China, Russia, and Mongolia. Social stability and economic development in border ethnic minority areas are key considerations of every policy. Therefore, it is necessary to study the impact of the PSRGEP on the local income gap.

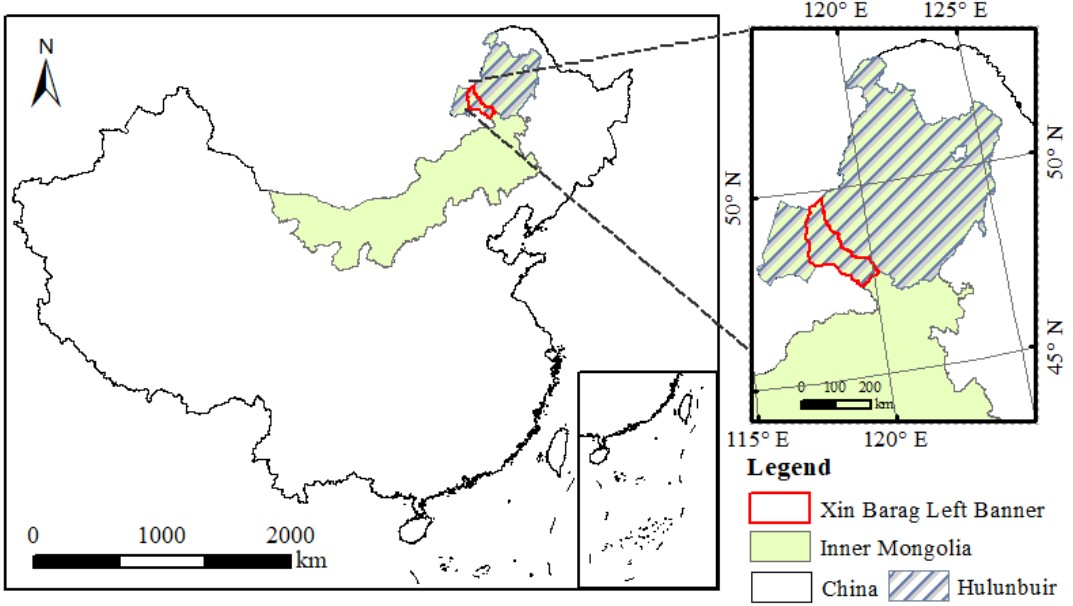

**Figure 1.** Location of Xin Barag Left Banner.

### 2.2. Data Acquisition

The research team conducted the study in Xin Barag Left Banner, Inner Mongolia Autonomous Region from April to May 2020. The total population of the whole banner is 41,813, of which 16,988 are engaged in animal husbandry. With the help of the local government and guides, we randomly surveyed herder households and obtained 203 valid questionnaires. The sample households covered every town in the whole banner (Amugulang Town, Cuogang Town, Ganzhuer Sumu, New Baolige Sumu, Wubuerbaolige Sumu, Jibuhulangtu Sumu and Handagai Sumu, a total of 7 towns/sumus). The interviewees were from 39 village-level administrative units including Wuerxun Community, Xilin Community, Ulantu Gacha, Hoh Wendu Gacha, etc. The random sampling method was chosen to make the samples more representative and avoid subjectivity, and so that our analysis would reveal the overall local situation as objectively as possible. The descriptive statistics (Table 1) show that the samples cover enough herder households of each income

level and grassland holding level required for our analysis. Additionally, according to what we learned in the interviews, the distribution of these two important indicators is similar to the overall local distribution. The survey content mainly includes the basic characteristics of the head of households (gender, age, ethnicity, type of employment, length of employment, etc. and detailed information for each type of capital can be found in Section 2.3), the current status of the five types of livelihood capital (natural capital, physical capital, financial capital, human capital, and social capital), and the benefits to herder households in regard to the PSRGEP (amount of compensation funds, production training experience, changes in living standards, etc.). During the investigation, we hired translators to facilitate our communication with local minority herdsmen. The duration of each survey questionnaire was approximately 50 min.

**Table 1.** Basic characteristics of sample herder households.

| Indicator | Category | Number of Samples | Proportion (%) |
|---|---|---|---|
| Gender of head of household | Male | 175 | 86.2 |
| | Female | 28 | 13.8 |
| Family's highest degree | Primary school | 13 | 6.4 |
| | Junior high school | 76 | 37.4 |
| | High school and above | 114 | 56.2 |
| Income level (yuan/a) | Less than 100,000 | 88 | 43.3 |
| | 100,000–200,000 | 53 | 26.1 |
| | 200,000–300,000 | 28 | 13.8 |
| | Above 300,000 | 34 | 16.7 |
| Main type of employment | Animal husbandry | 172 | 84.7 |
| | Crop farming | 1 | 0.5 |
| | Retail | 7 | 3.4 |
| | Migrant work | 15 | 7.4 |
| | Other | 8 | 3.9 |
| Holding grass area (ha) | Less than 200 | 83 | 40.9 |
| | 200–400 | 64 | 31.5 |
| | 400–600 | 35 | 17.2 |
| | Above 600 | 21 | 10.3 |

*2.3. Processing and Analysis*

First, we drew the Lorenz curve of the PSRGEP's fund allocation among the sample herder households. Then, we used the Gini index, which is widely used in current economics research, to describe the imbalance of fund distribution [37,38]. Then, the integrated livelihood capital scores before compensation were used as the baseline. The changes in ranking and standard deviation of the scores after receiving compensation funds were analysed to draw a conclusion about the impact on the income gap between herder households. Finally, we described the income gap through group comparison.

According to the sustainable livelihood approach (SLA) [39,40] proposed by the Department for International Development (DFID), combined with the livelihood characteristics of Xin Barag Left Banner herder households, this study developed a livelihood capital indicator system for sample herder households (Table 2). Financial capital is a source of income for herder households in addition to compensation funds. This represents the income level of herder households. Physical capital includes the herder households' houses, livestock, and durable goods and holdings. It represents the asset level of herder households in addition to income. These two kinds of capital can be quantified based on market value, and they are the main basis for measuring the income gap in Section 3.2.

**Table 2.** Livelihood capital and index weight of herders.

| Capital Type | Capital Weight | Indicator | Indicator Weight |
|---|---|---|---|
| Financial capital | 0.33 | Livestock income | - |
| | | Employment income | - |
| | | Retail revenue | - |
| | | Other subsidies | - |
| Physical capital | 0.26 | Livestock | - |
| | | Durable goods | - |
| | | Housing | - |
| Natural capital | 0.21 | Grazing prohibition grassland | 0.54 |
| | | Forage balance grassland | 0.46 |
| Social capital | 0.16 | Social participation | 0.10 |
| | | Number of relatives around | 0.59 |
| | | Reciprocity | 0.27 |
| | | Villager relations | 0.04 |
| Human capital | 0.04 | Number of laborers | 0.22 |
| | | Education | 0.54 |
| | | Health | 0.24 |

Natural capital is the household's grassland holdings. As the first means of production for herder households, grassland represents their production endowment. Social capital, including the social participation of herder households and interactions with relatives and friends around them, represents herder households' social connections. Human capital includes the herder households' family labour force, education level and health and represents the herder households' labour level. Since the meanings and dimensions of natural capital, social capital, and human capital are different, we standardized the values of the indicators for these three capital types from 0–1.

$$x_{ij} = \frac{a_{ij} - a_{min}}{a_{max} - a_{min}} \tag{1}$$

where $x_{ij}$ is the normalized value of $a_{ij}$ and $a_{ij}$ is the original value of index $j$ of household $i$. $a_{max}$ and $a_{min}$ represent the maximum and minimum values of $a_{ij}$, respectively. The entropy method was used to calculate the weight of each indicator for each livelihood capital type. The weight of index $j$ of household $i$ in the total number of indexes is:

$$P_{ij} = \frac{X_{ij}}{\sum_{i=1}^{n} X_{ij}} \tag{2}$$

where $P_{ij}$ is the weight of index $j$ of household $i$ in the total number of index $j$. Then, the entropy value of index $j$ can be obtained by the following formula:

$$e_j = -k \sum_{i=1}^{n} P_{ij} ln(P_{ij}) \tag{3}$$

where $e_j$ is the entropy of index $j$. Generally, $k = 1/ln\ n$, and $n$ is the number of sample herder households. The value of $e_j$ is positive. The weight of index $j$ in its capital is:

$$W_j = \frac{1 - e_j}{\sum_{j=1}^{m} (1 - e_j)} \tag{4}$$

where $W_j$ is the weight of index $j$ in its capital, and m is the total number of indicators included in the capital. Finally, the comprehensive score of the capital of household $i$ is:

$$s_i = \sum_{j=1}^{m} w_j p_{ij} \tag{5}$$

where $s_i$ is the comprehensive score of the capital of household $i$. Through the above methods, the comprehensive scores of natural capital, human capital and social capital were obtained. Then, we standardized the comprehensive scores of the five capital types from 0–1.

$$r_{iq} = \frac{s_{iq} - s_{min}}{s_{max} - s_{min}} \tag{6}$$

where $r_{iq}$ is the normalized value of $s_{iq}$; $s_{iq}$ is the score of capital $q$ of household $i$; and $s_{max}$ and $s_{min}$ represent the maximum and minimum values of $s_{iq}$, respectively. Then, we used the entropy method to determine the respective weights of the 5 types of capital $w_q$. The overall score of livelihood capital was obtained as follows:

$$R_i = \sum_{q=1}^{5} w_q r_{qi} \tag{7}$$

where $R_i$ is the overall score of the livelihood capital of household $i$.

## 3. Results

### 3.1. Unbalanced Distribution of Compensation Funds

We drew the Lorenz curve based on the allocation of compensation funds (Figure 2). The calculation results show that the Gini index (the ratio of the shaded area to the area of triangle OAB) of the sample households' compensation fund allocation is 0.46. This shows that the allocation of compensation funds for the policy is at a relatively unbalanced level [41,42]. Compensation funds from the PSRGEP are allocated according to the herder households' holdings of no-grazing pasture and forage–livestock balance pasture. Xin Barag Left Banner, Inner Mongolia Autonomous Region's standard subsidy award sis 205.25 yuan/ha per year for no-grazing pastures and 68.7 yuan/ha per year for forage–livestock balance pastures. Due to differences in the amount of grassland held by each household, the amount of compensation funds received also differs. We sorted the amount of compensation funds from highest to lowest. The highest compensation fund among the sample herder households was 215,410 yuan, and the lowest was only 229 yuan. According to the order of compensation funds from high to low, the top 20% of herder households received 49% of the total funds. The top 50% of herder households received 83% of the total funds. The lower 20% of herder households received only 3% of the total funds. Thus, the distribution was found to be unbalanced.

### 3.2. Impacts on Income Gap

Based on the unbalanced allocation of compensation funds, we used the trend lines (linear correlation analysis) to find the trend in the relationship between the herder households' livelihood capital and the compensation funds received (Figure 3a). The results show that the integrated score of herder households' livelihood capital is positively correlated with the amount of compensation funds received. In other words, herder households that receive more compensation funds tend to have higher overall livelihood capital scores. Above, we used the entropy method to obtain the weight of the herder households' five livelihood capitals. The herder households' physical capital (26%), financial capital (33%) and natural capital (21%) account for 80% of their overall livelihood capital score, and compared with human capital and social capital, these three types of capital can better reflect the economic level of local households. Considering the possible correlation between them, we used linear correlation analysis to fit the average score of the three major

types of capital to the compensation funds obtained (Figure 3b). The results show that the average score of the three major types of capital is positively related to the amount of compensation funds received by herder households. In other words, households who receive more compensation funds tend to have larger grassland areas, higher income, or more assets.

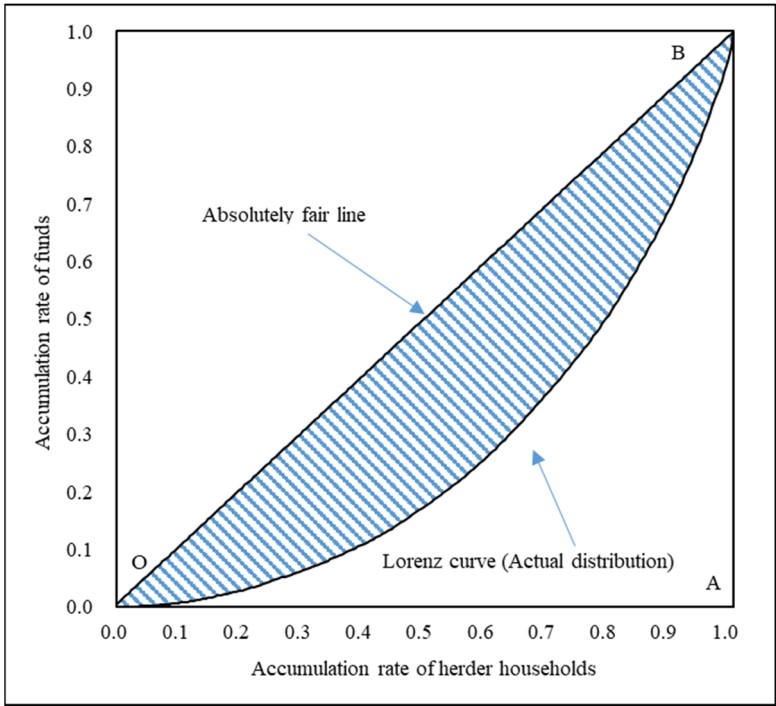

**Figure 2.** Lorenz curve of compensation fund allocation.

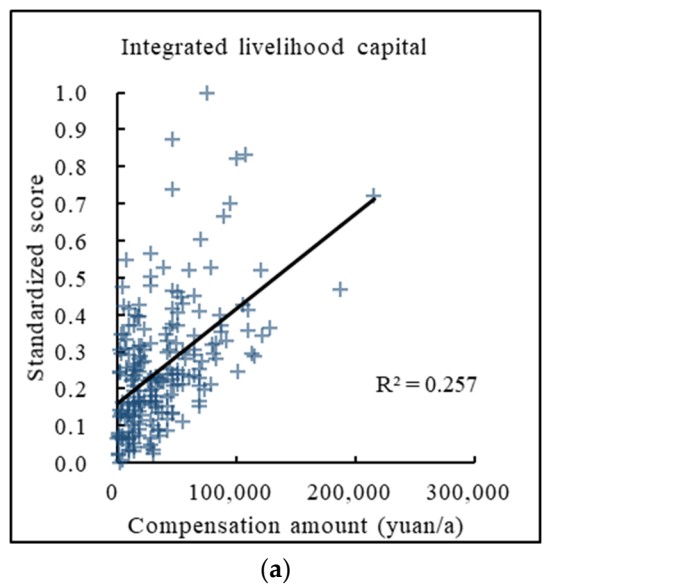

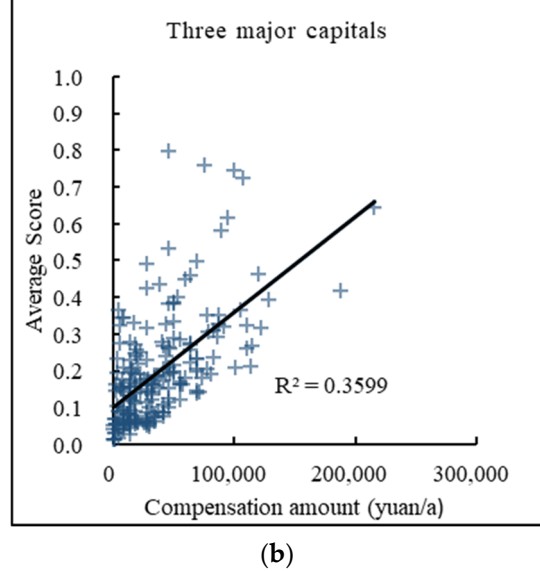

**Figure 3.** (**a**) The trend in the relationship between herder households' integrated livelihood capital and compensation amount; and (**b**) between herder households' three major types of livelihood capital and compensation amount. (scores ∈ (0, 1], Compensation amounts ∈ (229, 215, 410)).

After the sample herder households received the compensation funds, the ranking of their integrated livelihood capital score changed. However, due to the positive correlation between the households' integrated livelihood capital score and compensation funds, the

degree of change was small (Figure 4). The average absolute value of the changes in the ranking was only 4.87. The maximum change in ranking was 21, which is 10.3% of the total sample. There are 82 households whose absolute value of ranking change was less than 2, which is 40.4% of the total sample. The ranking of households whose integrated livelihood capital was at a medium level changed relatively drastically, while the change in the ranking of households whose integrated livelihood capital was at a low or high level was relatively small. After receiving the compensation funds, the standard deviation of the households' livelihood capital integrated scores rose from 0.1697 to 0.1734. The Gini index of the households' capital integrated scores decreased from 0.35 to 0.34 (the coefficient of variation decreased from 0.66 to 0.63). In summary, the absolute income gap between households has widened, but the relative income gap between households has been reduced.

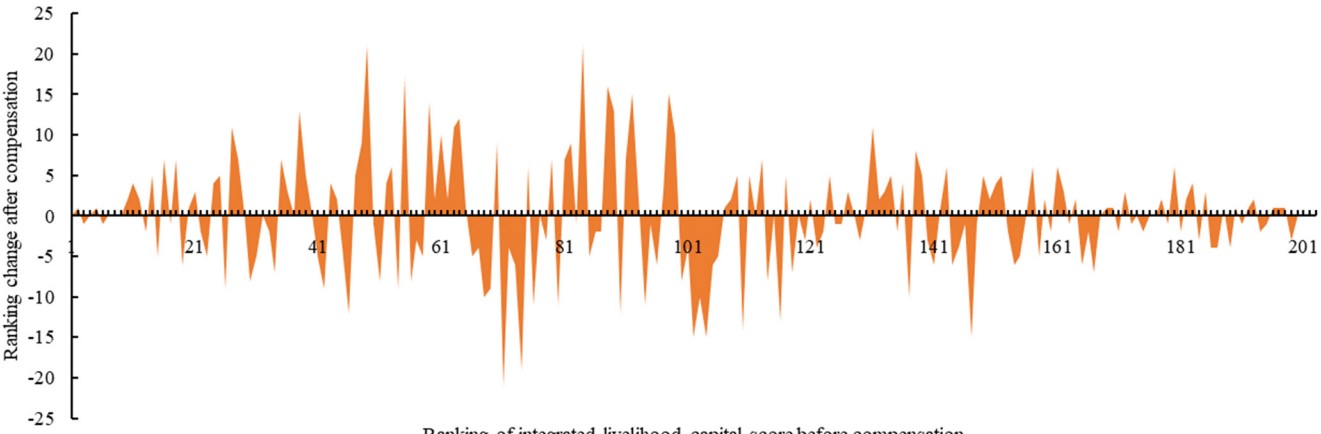

**Figure 4.** Changes in the ranking of integrated livelihood capital scores households before and after compensation.

Finally, we described the absolute income gap through group comparison. We sorted the integrated livelihood capital, financial capital, natural capital, and physical capital of the 203 herder households from high to low respectively. Then, they were divided into five groups (41-41-41-40-40). For the integrated livelihood capital (Figure 5a), the average compensation fund received by the group with the highest (59,625 yuan/a) was 3.6 times that of the group with the lowest score (16,667 yuan/a). For physical capital (Figure 5b), the average compensation funds obtained by the middle three groups were similar (39,885 yuan/a, 38,904 yuan/a and 36,472 yuan/a). The group with the highest physical capital received an average compensation fund of 50,137 yuan/a, which is 2.4 times that of the lowest group (20,740 yuan/a). For financial capital (Figure 5c), the groups with the highest financial capital (44,828 yuan/a) and the second highest (43,203 yuan/a) received similar compensation funds. The fourth group (36,987 yuan) received slightly more than the third group (35,689 yuan/a). However, the lowest group still received the smallest average compensation fund (25,562 yuan/a), which was far less than that received by the other four groups. For natural capital (Figure 5d), the group with the highest natural capital received nearly half of the total compensation funds, while the group with the lowest natural capital received 3.0% of the total compensation funds. In summary, the PSRGEP has indeed widened the income gap between herder households, and this gap is mainly reflected as polarization.

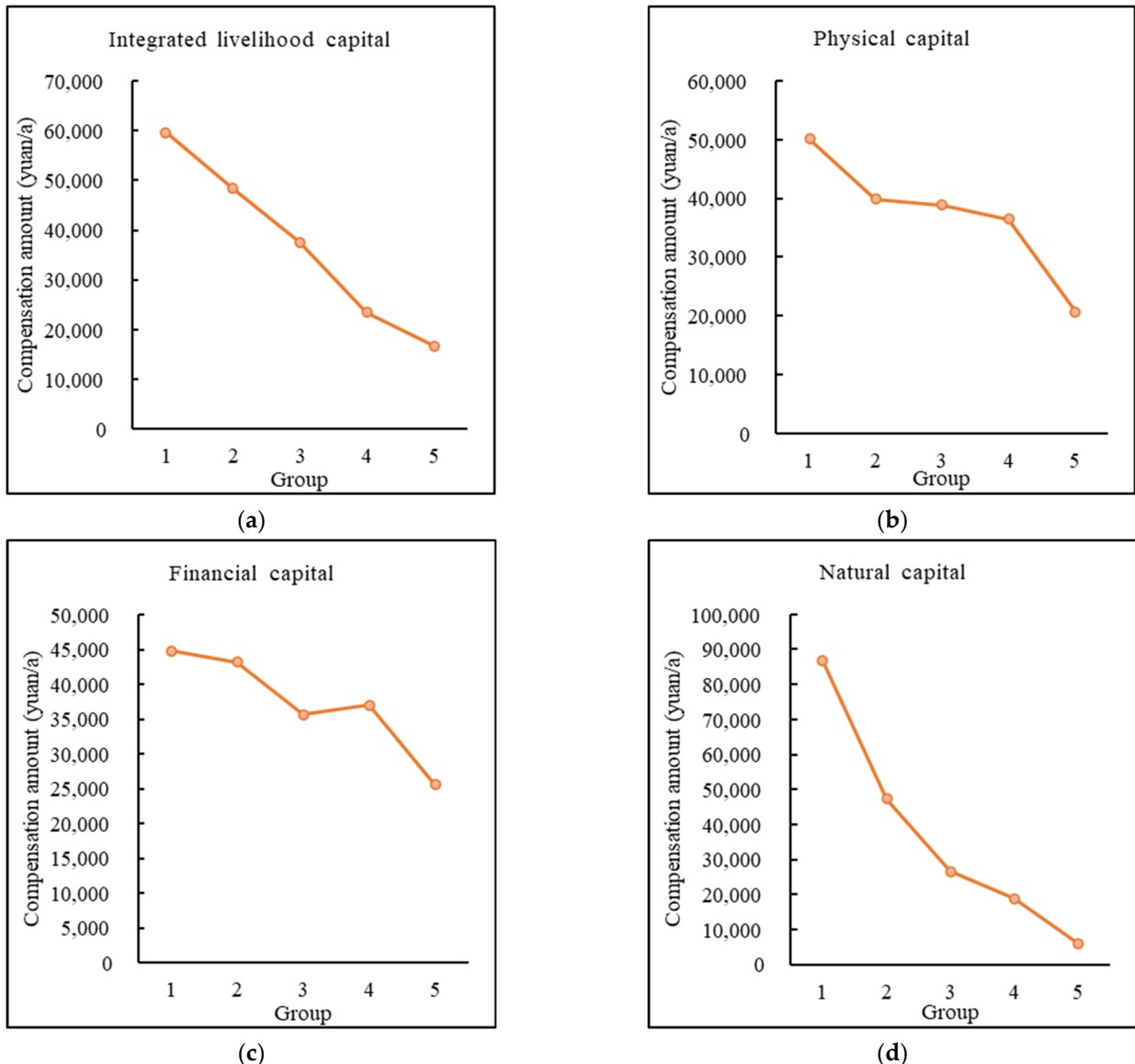

**Figure 5.** (**a**) Group comparison of herder households' compensation amount based on integrated livelihood capital. (**b**) Group comparison of herder households' compensation amount based on physical capital. (**c**) Group comparison of herder households' compensation amount based on financial capital. (**d**) Group comparison of herder households' compensation amount based on natural capital.

## 4. Discussion

Based on a survey of 203 herder households in Xin Barag Left Banner, Inner Mongolia Autonomous Region, this study explores the impact of PSRGEP on the income gap between herder households. The results show that the distribution of compensation funds is unbalanced (the Gini index is 0.46). According to the order of compensation funds from high to low, the top 20% of sample herder households received 49% of the total funds. Given the unbalanced distribution, households with better family economic conditions received more compensation funds. After receiving the compensation funds, the change in the ranking of household's livelihood capital integrated score was small, but the standard deviation rose from 0.1697 to 0.1734. The group with the highest integrated livelihood capital score received 3.6 times the compensation funds of the group with the lowest score. The Gini index of the households' capital integrated scores decreased from 0.35 to 0.34 (the coefficient of variation decreased from 0.66 to 0.63). As a result, under the promotion

of PSRGEP, the local absolute income gap has widened, but the relative income gap has reduced.

There have been several related studies on the impact of PSRGEP on herder households' income [28,31]. Most of these studies focused on the macro-evaluation of the policy's income-increasing effects. They described the impact of the policy on the overall income level of a certain area. However, this study focuses on the impact of PSRGEP on the income gap among herder households. It is innovative because it makes a horizontal comparison between micro-households.

As for income gap, many scholars have studied the income gap and its changes around the world [43–46]. These include absolute changes and relative changes. In this study, the changes in income gap focused more on the absolute change. There are three reasons for choosing the absolute change. First, for a policy measure that allows every household to receive funds, the change in absolute income gap can better reflect the rationality of the funds' distribution. Second, as mentioned in the introduction, the function of the policy is poverty alleviation, thus the change in absolute income gap can better answer whether the compensation funds were actually oriented towards poor people. The third and most important point, the purpose of the government is to provide compensation funds to encourage herder households to reduce livestock Small and medium-sized households are the main cause of concern in regard to overloading [47,48]. It is difficult for them to quantify the inherent economic inequality with large-scale households; however, the amount of compensation funds received by every household is relatively public. Therefore, a change in the absolute income gap is more likely to affect herders' enthusiasm for reducing livestock. This will have a direct impact on the effectiveness of the policy. Based on the above, we focused on the analysis of absolute income gap change.

The survey results of sample households in Xin Barag Left Banner show that the PSRGEP has widened the absolute income gap between herder households. However, according to the policy assumptions, the PSRGEP's funds are compensation based on the loss of opportunities for herder households to participate in the no-grazing and forage–livestock balance programs. Therefore, the households who received more compensation funds should have incurred greater losses. In that case, why did the income gap widen? Based on this investigation, we speculate that this is mainly due to the following reasons.

1.　Overloading is not effectively restricted

Only 35.0% of the sampled households had fewer livestock than they had in 2010 (the policy started in 2011). For most households (65.0%), the policy did not prompt them to reduce the number of livestock. This is due to many reasons. For example, some herder households were not overloading and did not need to reduce their livestock. Others did not know that they were overloading and did not reduce their livestock. Some herders knew that they were overloading, but due to the lack of strict supervision, they did not reduce livestock as required. In any case, based on the above facts, some herder households received compensation funds but did not reduce the number of livestock as required by the policy. In this way, "compensation" becomes a "donation" or even an "investment." If herder households with more pastures use more compensation funds to expand production, the absolute income gap will inevitably widen.

2.　Grassland transfer

A total of 47.0% of the herder households in the sample participated in grassland transfer (rented or leased). In interviews with local herder households, we learned that pastures were rented out by relatively affluent herder households to herder households that lacked sufficient pastures. However, according to the regulations, compensation funds are distributed only to the owners of the grassland. This means that the grassland owner who obtains the money does not bear any opportunity loss, while the renter who bears the opportunity loss is not compensated. This is another reason for the income gap.

Since the policy has such drawbacks, how can the policy be improved to reduce the income gap? We think the following three paths might be useful, because they aim to avoid the polarization of fund allocation and solve the two problems outlined above.

1.　Setting limits on the allocation of funds

From the perspective of fund allocation, the limit value of compensation can be formulated in the policy design. When the household's pasture holdings exceed a certain range, the compensation for the excess can be appropriately reduced. When the household's pasture holdings are less than a certain range, the compensation should be appropriately increased. In this way, the polarization caused by allocation of funds can be avoided.

2.　Strengthening policy supervision

The government should strengthen their supervision of the policy so that the herder households receiving compensation funds must control the number of livestock in accordance with the policy requirements. When the compensation funds serve only to compensate for the loss of opportunities for households, additional income gaps can be avoided. The policy will be fairer.

3.　Strengthening policy targeting

The current PSRGEP compensation method is haphazard and lacks accurate definitions for compensation types, compensation objects and compensation standards. The government should conduct a comprehensive investigation and accurately delineate the areas of no-grazing or forage–livestock balance. Compensation funds should be allocated to those who truly need compensation. The amount of compensation funds should match the opportunity loss of the recipient.

**Author Contributions:** Conceptualization, M.L.; methodology, M.L.; software, D.R.; formal analysis, Z.L.; investigation, Z.L.; resources, M.L.; data curation, D.R.; writing—original draft preparation, Z.L.; writing—review and editing, M.L.; supervision, M.L.; project administration, Z.L.; funding acquisition, M.L. All authors have read and agreed to the published version of the manuscript.

**Funding:** This work was jointly supported by the Mobility Programme DFG-NSFC (grant M-0342) and the National Natural Science Foundation of China (grant 42171279).

**Institutional Review Board Statement:** Not applicable.

**Informed Consent Statement:** Not applicable.

**Data Availability Statement:** The data presented in this paper are available on request from the corresponding author.

**Acknowledgments:** The authors wish to thank the government staff of New Barag Left Banner for their help with the questionnaire survey.

**Conflicts of Interest:** Authors would hereby like to declare that there are no conflict of interest in regard to the study.

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
