# Peer review of "The Impact of China’s Grassland Ecological Compensation Policy on the Income Gap between Herder Households? A Case Study from a Typical Pilot Area"

_land, doi:10.3390/land10121405_

Round 1
Reviewer 1 Report
The topic proposed by your manuscript is of high relevance for agricultural policy development in China. While the scope of the paper is regional, focusing on the performance and outcomes of the agricultural policy measures, it is of interest to the wider public, taking into account the global challenge of tackling the agricultural development and nature preservation interatctions.
in the text there are still some possibilities for improvement.
p4l135: ... the income gap in the following article. Please explain what is meant with the: following article. did you meant in the continuation of the article or something else?
p10l294: The content of the conclusions are not really the conclusions by a very short abstract. Please present the conclusions based on your results and eventual recommendations for the future measures of the China's agricultural policy or merge the chapters Discussion and Conclusions in one chapter.
Reviewer 2 Report
General Remarks
The topic is very relevant and interesting. It is well explained and is easy to read and understand. It deserves attention because it offers the opportunity for a discussion that has several different edges. However, I am somewhat disappointed for what I consider a short view of the important implications that the results may have. In addition, I have some doubts and concerns about the procedures and interpretation of the results. I’m not sure if the results presented in the manuscript allow for the interpretation claimed by the authors. My interpretation of the results, as a reader, does not coincide with them, nor with their conclusions and policy recommendations.
About the methods used for interpreting the results
One of the research questions highlighted by the authors in rows 75-76 is: Does the PSRGEP also widen the income gap between herder households? It is clear and perfectly natural that an income gap already existed among the herder households prior to the PSRGEP funds allocation (baseline situation) because all they had different livelihood capital endowments, as it was shown by the study. Is it clear, also, according to the way funds were to be allocated, that one should not expect a reduction of this gap by the sole application of the PSRGEP. As explained in row 37, the objective of this subsidy program is to reward or compensate herders for ecologically protecting grasslands.
Compensation funds from the PSRGEP, explain the authors in rows 172-173, are allocated according to the herder households’ holdings of no-grazing pasture and forage-livestock balance pasture. Thus, those herder households holding more grass areas would be the ones receiving more funds, just by definition. It would be perfectly possible that, in absolute terms, the income gap could even increase. Of course, the real question would be to check if the net result tended to be greater in relative terms. In my opinion, just saying that the Gini index of the households’ compensation fund allocation is 0.46 tells nothing other than confirming that the funds are not evenly distributed. What makes an index of 0.4 be a warning line (row 170)? In 2016, the World Bank estimated the Gini Index for the whole of China as 0.385 (https://www.indexmundi.com/ china/ distribution_of_family_income_gini_index.html). We know that these estimates change. What is the magic number?
In any case, I think it would be better to set a baseline, draw a Lorenz curve and calculating the corresponding Gini index before and after the allocation, and statistically compare the results. In addition, by comparing the ranking of households by their overall livelihood capital before and after the PSRGEP intervention it would be possible to observe if there are changes in the relative positions of the herder households. I think that this procedure could give a better answer about if the income gap between herder households has widened because of the PSRGEP.
To somewhat answer this question, the authors use a linear regression model to fit the trend between the herder households’ livelihood capital and the compensation funds received. First, the apparent simplicity of estimating a simple linear regression model is not a justification for avoiding some further explanation about the procedure. It seems to me that we should find heteroskedastic errors by construction, and I would like to be sure the authors tested and corrected this issue if found. Second, what is the measure of the positive relationship? As shown in Figure 3 (row 206), the variable “compensation amount” is limited to non-negative values, which may introduce bias in the estimation of the regression coefficients. Nothing is said about this. Third, assuming that the regression estimates are ok, from the regression results the authors contend that “the overall score of herder households’ livelihood capital is positively correlated with the money value of compensation funds received. In other words, herder households who receive more compensation funds tend to have higher overall livelihood capital score” (rows 189-192). Ok, I may agree on that but, by no means I can conclude this is proof that the income gap between herder households has widened in relative terms. The last step regarding the grouping of the 203 herder households into the five groups correctly depicts (if I understand, because is not explicated, that the order 1 to 5 goes from richer to poorer households) the positive relationship between “welfare” (measured through the livelihood capital and its three main components, physical, financial, and natural) and the allocation of funds, knowing that this is also related to the allocation rules, as the authors recognize in rows 226 and 227 (“the natural capital score directly determines the compensation funds to a certain extent”). However, it does not constitute, in my view, proof of the widening of the income gap.
About the discussion and conclusions
The concerns stated previously do not pretend to say that, in fact, the income gap was reduced or remained unaltered after the implementation of the PSRGEP in Xing Barang Left Banner. I only say I cannot infer that only from the results exposed in the manuscript. It perfectly could be the case, though, that effectively the income gap widened. In the Discussion section, the authors present a few plausible reasons for this widening really occur:
- Overloading was not effectively restricted
- Grassland transfers by rent or lease
Regarding #1., the authors contend that if by not reducing the number of grazing animals, the pressure over the grasslands production may not decrease, and could even increase, this could lead the policy to have negative effects (rows 261-263). In a certain way, this could be exacerbated with reason #2, that is, the fact that compensation funds to be distributed only (I read, exclusively) introduces an additional bias against “poorer” households. The problem is, what are the implications or how to interpret these issues. The authors consider some policy improvements to narrow the income gap:
- Setting limits on the allocation of funds
- Strengthening policy supervision
- Strengthening policy targets
The solutions proposed by the authors lie outside the research evidence provided by the study. They are completely normative, and they are not followed by any further analysis. Their interpretation circumscribes on the lack of control. It seems that the policy is good, maybe the implementation was wrong or had defects. Is it really the problem or can one argue that, maybe, the whole policy itself is wrong and produces unwanted effects? In the introduction, the authors cited a couple of works ([25,26]) stating that “since the implementation of the policy, the quality of grassland in China has improved significantly” and took this for good. However, looking at their own results, they argue that the degradation of grasslands is getting worse (again rows 261-263).
Considering the proposal of limiting the allocation of funds and strengthening policy targets and policy supervision, one may ask why these actions are needed? Is it because households have an intrinsic willingness to cheat? Or, on the contrary, there is a lack of sense in the policies. Why are so sure the authors that with these measures, the policy might work?
About point (a) they say “When the pasture holdings of households exceed a certain range, the compensation for the excess can be appropriately reduced. When the households' pasture holdings are less than a certain range, the compensation should be appropriately increased. In this way, the polarization caused by the allocation of funds can be avoided” (rows 276-279). It seems easy, right? What is an appropriate and objective range? What are the upper and lower limits?
Regarding point (b), “When the compensation funds serve only to compensate for the loss of opportunities for households, additional income gaps can be avoided. The policy will be fairer” (rows 284-286). Will stronger policy supervision ensure that the compensation fund strictly serves to compensate for the opportunity loss for households? How is that? How much cost does this solution imply? In the case under study, we are talking about 250 million yuan allocated to protect 1.54 million hectares. This is a very small part of China’s grasslands (30-40% of the country´s land area). That means slightly more than 100 yuan/ha. In US dollars, we are talking of near $23.5 million, meaning about $/ha 15. What would be the total additional costs of more supervision at the national level?
Point (c) proposes to conduct a comprehensive government investigation to accurately delineate the pastures of no-grazing or forage-livestock balance, to help those who truly needed compensation (rows 290-292). By what means can one end up with an identification and selection procedure that objectively ensures that. If any, what would be the costs, again, considering implementing this at the national level.
The objections posed to the discussion of the results are not a motive for rejecting nor conditioning the acceptance of the manuscript. On the contrary. If the authors find a way to overcome the critics on the methods used for interpreting the results, even if maintaining the discussion and conclusions I will be happy to read this manuscript published because it raises an interesting discussion and invites for a good confrontation of ideas.
Reviewer 3 Report
This manuscript is well structured and written with clear objectives. However, there are some shortcomings that need to be addressed before the manuscript can be accepted for publication.
There is limited information on the data acquisition in the methods section. More information on the total population of the study area, the name and number of villages and more details about the social survey need to be added.
Why was random sampling method used? How did this method help the authors recruit a representative sample of household herders in the study area?
The results/findings have not been generalized to be useful for future research. So, it would be good if the authors discuss about the key findings in relation to other studies’ findings conducted based on similar approaches in different places across the world. Broader implications of the findings should also be stated at the end of the abstract.
